# The Diagnostic and Therapeutic Role of Circular RNA HIPK3 in Human Diseases

**DOI:** 10.3390/diagnostics12102469

**Published:** 2022-10-12

**Authors:** Yanfei Feng, Zitong Yang, Bodong Lv, Xin Xu

**Affiliations:** 1The Second Clinical Medical College, Zhejiang Chinese Medical University, Hangzhou 310053, China; 2Department of Urology, The Fourth Affiliated Hospital, Zhejiang University; Hangzhou 322000, China; 3Department of Urology, The Second Affiliated Hospital, Zhejiang University; Hangzhou 310009, China; 4Department of Urology, First Affiliated Hospital, School of Medicine, Zhejiang University, Hangzhou 310003, China

**Keywords:** circRNAs, circHIPK3, human diseases

## Abstract

Circular RNAs (circRNAs) are a class of noncoding RNAs with closed-loop of single-stranded RNA structure. Although most of the circRNAs do not directly encode proteins, emerging evidence suggests that circRNAs play a pivotal and complex role in multiple biological processes by regulating gene expression. As one of the most popular circRNAs, circular homeodomain-interacting protein kinase 3 (circHIPK3) has frequently gained the interest of researchers in recent years. Accumulating studies have demonstrated the significant impacts on the occurrence and development of multiple human diseases including cancers, cardiovascular diseases, diabetes mellitus, inflammatory diseases, and others. The present review aims to provide a detailed description of the functions of circHIPK3 and comprehensively overview the diagnostic and therapeutic value of circHIPK3 in these certain diseases.

## 1. Introduction

During the past few decades, an enormous number of studies have outlined the complex regulatory networks of various RNAs in multiple human diseases [1]. Generally, according to whether they encode proteins or not, RNAs are divided into two types—protein-encoding messenger RNAs (mRNAs) and noncoding RNAs. Noncoding RNAs, as the name indicates, cannot directly encode proteins, yet account for approximately 98% of the total transcriptome and play a pivotal role in gene expression [2]. In recent years, the understanding of noncoding RNAs has gradually deepened, with the development of sequencing technologies and bioinformatics [3], notably insights into the mechanisms of different noncoding RNAs in cell biology.

Circular RNAs (circRNAs), one of the hottest noncoding RNAs currently studied, exist in the form of a continuous closed loop. Initially, they were mistaken as meaningless byproducts of RNA splicing. Nonetheless, recent studies have shown that their significance is underestimated. First, advanced sequencing and bioinformatics tools have uncovered the fact that circRNAs are endogenously abundant [4]. Distinguished with linear RNAs, circRNAs have a more conservative and stable structure, with neither a 5′ cap nor a 3′ poly A tail, making these molecules more tolerable to degradation by exonucleases [5]. Evidence indicates that the majority of circRNAs are present in the cytoplasm of eukaryotic cells. Moreover, the widespread but discrepant expression of circRNAs in various organs and cells suggests that they may participate in multiple crucial physiological and pathological processes [6,7].

Actually, the regulatory functions of circRNAs in gene expression are diverse and complicated. In most cases, circRNAs bind specific microRNAs (miRNA) and act as competitive endogenous RNAs (ceRNAs) or so-called “miRNA sponges” [8]. In addition, circRNAs have the opposite abilities to act as miRNA reservoirs, stabling and activating some other miRNAs. Furthermore, circRNAs are reported to bear relevance to gene expression both upstream and downstream in three aspects. First, it has been reported that circRNAs may correlate with RNA polymerases to regulate the transcription complex. Second, another regulation pattern called “back-splicing” can be initiated by circRNAs to inhibit linear splicing. Third, circRNAs can interact with RNA-binding proteins (RBPs) to act as protein antagonists [9]. Counterintuitively, although it is conventionally believed that circRNAs are not equipped with encoding capabilities, some studies have illustrated that certain circRNAs might encode proteins if internal ribosome entry sites (IRESs) exist in their structures [10].

Circular homeodomain-interacting protein kinase 3 (circHIPK3), which originates from the second exon of the HIPK3 gene, is an eye-catching molecule among all the circRNAs [11]. Plenty of studies have demonstrated the biogenesis of circHIPK3 and its vital roles in the progression of multiple human diseases, including cancers, cardiovascular diseases, inflammatory diseases, diabetes mellitus, etc. [12,13]. This review aims to provide a more detailed description of the functions of circHIPK3 and a more comprehensive overview of its diagnostic and therapeutic values in some of the aforementioned diseases.

## 2. Biogenesis of circHIPK3

The HIPK3 gene is an important member of the nuclear kinases HIPK family, in addition to HIPK1, HIPK2, and HIPK4 [14], located on chromosome 11p13 of humans. The results of genomic sequencing reveal that circHIPK3 consists of exon 2 with 1099bq flanked on both sides by two long introns with highly complementary Alu repeats, which are considered as the key elements in the process of intron-pairing driven circularization [11,15]. The circularization of circHIPK3 was reported to be dependent on the canonical spliceosome mechanisms in HeLa cells, and the flanking introns could mediate the efficiency of circularization [16]. A study on Drosophila revealed that the biogenesis of some circRNAs, including circHIPK3, could be co-modulated by cis-acting elements and trans-acting splicing factors such as heterogeneous nuclear ribonucleoproteins (hnRNPs) and SR proteins [17]. Intriguingly, scientists illustrated that double-stranded RNA (dsRNA)-specific adenosine deaminase (ADAR) enzymes also play an important role in circRNA formation [18]. The regulation of circRNA production is complicated. Although the regulatory networks of the biogenesis of circRNAs have not been fully elucidated [19], it can be determined that circRNAs, including circHIPK3, are ubiquitous in a variety of tissues including brain, heart, lung, etc. [20]. Figure 1 depicts the biogenesis of circHIPK3.

## 3. The Role of circHIPK3 in Human Cancers

Accumulating studies have substantiated that the dysregulation of circHIPK3 contributes to multiple processes in the carcinogenesis and progression of various human cancers [21]. Not only does this molecule play a pivotal role in the proliferation, migration, invasion, and drug resistance of tumor cells, but also it has the potential to act as a biomarker in early diagnosis and prognosis [22], considering that in most cases, circHIPK3 is a tumor promoter by sponging miRNAs. Table 1 displays the expression status, the regulatory targets, and the associated functions of circHIPK3 in human cancers.

### 3.1. Lung Cancer (LC)

As one of the most life-threatening human malignancies, lung cancer has been a global challenge during the past years [23]. Several published reports have provided us some novel clues of the occurrence and progression of LC mediated by overexpressed circHIPK3. Yu and colleagues elucidated that enriching circHIPK3 facilitated the expression of sphingosine kinase 1 (SphK1), cyclin-dependent kinase 4 (CDK4), and signal transducer and activator of transcription 3 (STAT3) through impeding miR-124 in lung cancer cells, and accelerating LC cell growth [24]. Chen et al. also illustrated another signaling pathway mediated by circHIPK3/miR-124-3p. STAT3 was reported to be the downstream target of miR-124-3p in this case [25]. Lu and others explored another miRNA which was involved in the carcinogenesis and progression of nonsmall-cell lung cancer (NSCLC) mediated by circHIPK3. Their research revealed that circHIPK3 negatively regulated miR-149, promoting forkhead box M1 (FOXM1) expression. Eventually, cell proliferation was triggered when apoptosis was blocked [26]. MiR-107 was introduced by Hong and colleagues indicating that circHIPK3 downregulated miR-107, reversing its inhibitive effect on brain-derived neurotrophic factor (BDNF), which facilitated NSCLC progression [27]. However, Zhao et al. observed a lower expression level of circHIPK3 in LC specimens compared with surrounding nontumor tissues. Moreover, patients with lower circHIPK3 were more likely to have larger tumor size, more advanced TNM stage, and more lymph node infiltration. Further experiments on gefitinb-resistant cell line revealed that low circHIPK3 contributed to gefitinib resistance [28].

### 3.2. Hepatocellular Carcinoma (HCC)

HCC is a common malignancy with high recurrence and mortality rate around the world [23]. An augmented level of circHIPK3 in HCC was validated in several studies. Chen and colleagues observed that the increased level of circHIPK3 was positively associated with Aquaporin 3 (AQP3) expression in HCC specimens. Mechanistically, miR-124 was considered as an intermediate link between circHIPK3 and AQP3. Intriguingly, the aberrant expression of circHIPK3 was closely correlated with several clinical parameters such as tumor stage, the amount of HBV–DNA copy and the existence of liver cirrhosis in patients with HCC [29]. MiR-124 was also confirmed as the target of circHIPK3 in other studies. Yu et al. observed that the upregulation of circHIPK3 was associated with depletion of miR-124 and miR-506. Pyruvate dehydrogenase kinase 2 (PDK2) was reported to be the downstream target of miR-124 or miR-506. The circHIPK3-miR-124/miR506-PDK2 signaling gave rise to carcinogenesis and progression of HCC cells [30]. Hu and others demonstrated the expression of miR-124-3p and miR-4524-5p was inhibited by circHIPK3, promoting multidrug-resistant protein 4 (MRP4) expression. Although their findings suggested MRP4 was not directly involved in occurrence and progression of HCC, MRP4 itself was proved to be aberrantly overexpressed in multiple cancers [31]. Another miRNA involved in the association between circHIP3 and HCC was miR-338-3p introduced by Li et al. Similarly, circHIPK3 served as the sponge of miR-338-3p, hindering zinc finger E-box binding homeobox 2 (ZEB2) expression. This signaling eventually led to epithelial–mesenchymal transition (EMT) and HCC metastasis [32].

### 3.3. Esophageal Squamous Cell Carcinoma (ESCC)

ESCC is a malignancy of the digestive system which severely affects quality of life [23]. Yao et al. reported that an elevated level of circHIKP3 in ESCC cells remarkably absorbed miR-599 expression and significantly stimulated c-MYC expression, leading to cell growth and metastasis [33]. Yao and colleagues validated that miR-124 functioned as the sponged miRNA in this case and serine/threonine kinase 3 (AKT3) was enhanced under the modulation of circHIPK3, which further triggered ESCC cell growth and metastasis [34].

### 3.4. Gastric Cancer (GC)

The incidence of GC has been gradually escalating during the past decades [23]. Most of the emerging studies demonstrated an overexpression of circHIPK3 in GC cells. Cheng et al. discovered that circHIPK3 was even higher in metastatic GC cells. They further unveiled the underlying mechanisms that circHIPK3 suppressed the expression of miR-29b and miR-124, enhancing GC cell replication. Furthermore, this signaling pathway was considered to be a prognostic factor of GC [35]. Wei et al. elucidated the connections between circHIPK3 (level, stage, grade) and the prognosis of GC. Knocking down circHIPK3 significantly extenuated GC progression by upregulating miR-107 and then depleted brain-derived neurotrophic factor (BDNF) [36]. Another miRNA introduced by Li and others was miR-876-5p. Their research showed that circHIPK3 sponged miR-876-5p, reversing its inhibitive effects on phosphoinositide-3-kinase regulatory subunit 1 (PIK3R1), an oncogene proved to exist in multiple tumors, including GC [37]. A study conducted by Jin et al. illustrated that circHIPK3 contributed to metastatic disease of GC by downregulating miR-653-5p and miR-338-3p. Neuropilin 1 (NRP1) was determined as the inhibited target downstream of these two molecules and its depletion stimulated extracellular signal-regulated kinase (ERK) and protein kinase B (AKT) pathways [38]. Yang et al. observed another regulatory pathway in which circHIPK3 suppressed miR-637, rescuing the function of serine/threonine kinase 1 (AKT1) sabotaged by miR-637 and then enhancing GC cell progressive activities [39]. Liu and colleagues observed that silencing circHIPK3 decreased GC cell proliferation and migration. The Wnt/β-catenin pathway was considered as the intermediate factor of this process [40]. However, Ghasemi and colleagues arrived at a contrary conclusion, that a dramatically decreased level of circHIPK3 in GC was proved to be associated with patient age and clinical stage [41]. The conflicting conclusions drawn by different scientists made the role of circHIPK3 in GC more complicated.

### 3.5. Colorectal Cancer (CRC)

CRC is another most common gastrointestinal cancer which hugely affects quality of life [23]. A study organized by Zeng et al. clarified that an augmented level of circHIPK3 in CRC cells blocked miR-7 expression and then contributed not only to the activation of cell proliferation and metastasis, but also inhibits cell apoptosis [42]. Yan et al. also discovered an increase in circHIPK3 level in CRC cells. Another miRNA was introduced by them, miR-1207-5p, which was negatively regulated by circHIPK3 and whose inhibitive effects on formin-like 2 (FMNL2) were relieved. This signaling pathway consequently resulted in CRC cell replication and metastasis [43]. Zhang et al. explored the oxaliplatin-resistant mechanisms of CRC and demonstrated that circHIPK3 acted as the sponge of miR-637 to enhance STAT3 expression, triggering Bcl-2/beclin1 afterward, and as a result detecting decreased autophagy. Additionally, they observed connections between an elevated level of circHIPK3 and clinical indicators of poor prognosis [44].

### 3.6. Prostate Cancer (PCa)

The incidence of PCa ranks first in men [23], in which an unregulated level of circHIPK3 was validated in several studies. In the research of Chen et al., circHIPK3 was identified as a crucial factor of prognosis. Via sponging miR-193a-3p, circHIPK3 promoted myeloid cell leukemia 1 (MCL1) expression, and activated PCa cell growth and tumor progression [45]. Cai and colleagues demonstrated that the enrichment of circHIPK3 stimulated the proliferation and invasion of PCa cells by impeding miR-338-3p and strengthening a disintegrin and metalloproteinases 17 (ADAM17) expression [46]. Another study by Liu et al. identified other downstream molecules of miR-338-3p as Cdc2 and Cdc25B and this signaling pathway modulated by circHIPK3 led to acceleration of G2/M transition and the proliferation of PCa cells [47]. Liu and others uncovered another pathway mediated by circHIPK3 in which circHIPK3 downregulated miR-448 to activate metadherin (MTDH), thus accelerating PCa cell proliferation and metastasis [48]. Tang et al. identified an overexpression of exosomal circHIPK3 in PCa, which suppressed exosomal miR-212 and relieved its negative modulation on B-cell-specific MMLV insertion site-1 (BMI-1), consequently activating cell growth and metastasis [49].

### 3.7. Renal Cell Carcinoma (RCC)

RCC is one of the urinary cancers with high prevalence [23]. Lai and colleagues detected that an elevated expression of circHIPK3 in RCC cells blocked the function performed by miR-485-3p on several molecules related to EMT and cell death. For example, the overexpression of clever caspase-3, Bax, and E-Cadherin was hampered and the decrease in Vimentin, N-Cadherin, and Bcl-2 was reversed [50]. Han et al. verified that by hindering expression of miR-508-3p, the increased level of circHIPK3 boosted RCC cell viability. The downstream target of miR-5083p was determined as chemokine ligand 13 (CXCL13), whose activation was also validated to be associated with several cancers [51]. Omata and others discovered an upstream regulator of circHIPK3 as adenosine deaminase acting on RNA 1 (ADAR1), which could inhibit circHIPK3 and then activate miR -381-3p, so as to block the translation process of MRP4 protein, a factor which was clarified to be related with drug resistance [52]. On the contrary, Li et al. elaborated that circHIPK3 was sharply diminished in RCC cells, and that lifting circHIPK3 expression repressed the invasiveness of RCC cells by hampering miR-637 [53].

### 3.8. Bladder Cancer

Bladder cancer is another frequently diagnosed cancer of the urinary system [23]. Compared with other malignancies, circHIPK3 tended to be downregulated in bladder cancer cells. Li et al. showed that circHIPK3 expression was negatively associated with clinical stage, invasiveness, and lymph node infiltration in bladder cancer patients. Mechanistically, circHIPK3 sponged miR-558 to hinder heparanase (HPSE) expression, reducing the development of bladder cancer cells [54]. Xie and colleagues clarified the prognostic value of circHIPK3 in bladder cancer and illustrated that an elevated expression of circHIPK3 enhanced the response to gemcitabine chemotherapy [55]. Remarkably, the analysis conducted by Okholm et al. demonstrated that circHIPK3 was rigidly related with aggressiveness in nonmuscle-invasive bladder cancer, further highlighting the prognostic role of circHIPK3 in bladder cancer [56].

### 3.9. Breast Cancer

Breast cancer has created onerous economic and social burden owing to its high prevalence in females [23]. Qi and colleagues along with Luo et al. detected an aberrantly elevated expression of circHIPK3 in breast cancer. The downstream target of circHIPK3 was considered to be miR-326, and the depletion of miR-326 mediated by circHIPK3 resulted in cell replication, migration, and invasion of breast cancer [57,58]. Shi et al. demonstrated that an upregulated circHIPK3 targeted miR-124-3p to boost MTDH expression, facilitating angiogenic activities of breast cancer cells [59]. Chen et al. identified miR-193a as another target of circHIPK3. Further, the signaling pathway went downstream to high mobility group box-1 (HMGB1)/phosphoinositide-3-kinase (PI3K)/AKT and the proliferative and metastatic abilities of breast cancer cells were enhanced as a consequence [60]. Zhang et al. offered deep insight into the mechanisms of trastuzumab resistance in breast cancer and discovered that exosomal circHIPK3 was capable of lowering the sensitivity of breast cancer cells to trastuzumab [61]. A study managed by Ni and colleagues revealed another phenomenon of chemoresistance in which circHIKP3 overexpression suppressed miR-1286 to enhance hexokinase 2 (HK2) expression and decreased the response to paclitaxel treatment in breast cancer cells [62].

### 3.10. Ovarian Cancer (OC)

OC is not uncommon in the female genital system [23]. Liu et al. uncovered that circHIPK3 expression was hugely increased in OC tissues and cell lines, which was considered to correlate with unfavorable outcomes of OC [63] as well. However, Teng and colleagues detected a reduction in circHIPK3 level in OC cells. CircHIPK3 depletion was validated to stimulate cell growth, invasion, and migration, on the one hand, and, on the other, prevent cell death. A number of miRNAs, such as miR-10a, miR-106a, and miR-148b, were assumed to be modulated in this process, but exactly which miRNAs were inhibited remained elusive [64].

### 3.11. Cervical Cancer (CC)

Cervical cancer, one of the most common genital cancers in postmenopausal women, usually causes poor outcomes [23]. Wu and colleagues observed a dramatically upregulated status of circHIPK3 in CC cells. The miRNA sponged by circHIPK3 was reported to be miR-485-3p, and its depletion accelerated fibroblast growth factor 2 (FGF2) expression, strengthening the tumorigenic and progressive abilities of CC cells [65]. Qian et al. validated miR-338-3p as another molecule inhibited by circHIPK3. Accordingly, hypoxia-inducible factor-1alpha (HIF-1α) activated and triggered EMT to stimulate CC cell proliferation, migration, and invasion [66].

### 3.12. Osteosarcoma

Osteosarcoma is a type of severe disease frequently diagnosed in youngsters [23]. Huang and colleagues demonstrated an augmented expression of circHIPK3 in osteosarcoma cells, which downregulated miR-637, relieved its negative regulation on STAT3, and caused osteosarcoma cell metastasis [67]. Similarly, Wen et al. introduced another signaling pathway in which circHIPK3 mediated osteosarcoma progression by decreasing miR-637 to facilitate histone deacetylase 4 (HDAC4) expression [68]. However, Ma et al. observed a strikingly decreased level of circHIPK3 in osteosarcoma cells and specimens, which correlated with an advanced stage, distant metastasis, and unfavorable prognosis. While a higher circHIPK3 expression dramatically declined aggressiveness of tumor cells [69], the contradictory conclusions obtained by these studies requires further exploration.

### 3.13. Glioma

With unfavorable prognosis, glioma is one of the most malignant tumors in the central nervous system [23]. Emerging published reports clarified an increased level of circHIPK3 in glioma cells. Jin et al. validated that circHIPK3 promoted insulin-like growth factor 2 mRNA-binding protein 3 (IGF2BP3) expression by blocking miR-654, which expanded the proliferative and metastatic capabilities of glioma cells [70]. Hu and colleagues demonstrated that a high expression level of circHIPK3 significantly suppressed miR-124-3p to rescue its negative effects on STAT3. This pathway had a promotive influence on tumorigenesis and progression of glioma cells [71]. Another target of circHIPK3/miR-124 interaction determined by Liu et al. was cyclin D2 (CCND2), which again spurred the development of glioma [72]. CircHIPK3 was also considered to be inextricably bound with temozolomide treatment of glioma, and two signaling pathways were introduced by scientists. One study of Han et al. verified that exosomal circHIPK3 reduced the sensitivity to temozolomide treatment via absorbing the effect of miR-421 and triggering zinc finger of the cerebellum 5 (ZIC5) expression [73]. The other study conducted by Yin and colleagues identified miR-524-5p as the direct target of circHIPK3 and kinesin family member 2A (KIF2A) the downstream molecule [74].

### 3.14. Oral Squamous Cell Carcinoma (OSCC)

OSCC is a particularly severe malignant tumor with poor outcomes that occurs in the regions of head and neck [23]. Bi et al. uncovered the enhanced expression of circHIPK3 in OSCC and clarified its further target molecule as miR-381-3p. Yes-associated protein1 (YAP1) was determined to be the downstream factor whose upregulation stimulated the carcinogenesis and progression of OSCC cells [75]. Similarly, the findings of Jiang et al. suggested that miR-637 was negatively modulated by augmented circHIPK3, triggering nuclear protein 1 (NUPR1) and PI3K/AKT pathway to activate OSCC cell growth and metastasis [76].

### 3.15. Leukemia

The incidence of leukemia has been increasing during the past years [23]. Feng et al. found that circHIPK3 was enormously overexpressed in the blood cells and serum of chronic myeloid leukemia (CML), suggesting a worse outcome [77]. Similarly, Gaffo and colleagues detected an abnormally increased expression of circHIPK3 in acute lymphoblastic leukemia (ALL) cells, which provided novel findings in this issue [78].

### 3.16. Other Cancers

Apart from those mentioned, there have been connections between circHIPK3 and other cancers including nasopharyngeal cancer (NPC), pancreatic cancer, thyroid cancer, gallbladder cancer (GBC), melanoma, and glioblastoma. Ke et al. revealed an augmented expression of circHIPK3 in NPC and a dramatically declined miR4288 expression. Mechanistically, circHIPK3 blocked miR4288 in order to boost the expression of E74-like ETS transcription factor 3 (ELF3), and promote NPC cell growth [79]. In terms of pancreatic cancer, Liu and colleagues demonstrated that the overexpression of circHIPK3 remarkably downregulated miR-330-5p-activated RAS-association domain family 1 (RASSF1) expression, stimulating the carcinogenesis and progression and becoming immune to gemcitabine therapy of pancreatic cancer cells [80]. Shu et al. showed an aberrantly increased circHIPK3 level in thyroid cancer cells and identified miR- 338-3p as the downstream sponged factor. Ras-like in rat brain 23 (RAB23) was enhanced under the modulation of circHIPK3, which contributed to tumorigenesis and progression of thyroid cancer cells [81]. Regarding GBC, circHIPK3 expression was found significantly higher in GBC cells by Kai and colleagues. MiR-124 was considered to be sharply suppressed by circHIPK3, escalating the expression of rho-associated protein kinase 1 (ROCK1) and cyclin-dependent kinase 6 (CDK6), which caused the accelerated growth of GBC cells [82]. Zhu et al. observed that circHIPK3 was remarkably overexpressed in melanoma cells and the downstream target molecule was confirmed to be miR-215-5p, the depletion of which significantly triggered Yin Yang 1 (YY1) expression and thereby expedited cell growth and mitigated cell death of melanoma [83]. Stella and colleagues discovered correlations between the reduction in circHIPK3 in serum extracellular vesicle and the prognosis of glioblastoma and suggested that circHIPK3 could possibly serve as a promising biomarker [84].

## 4. The Role of circHIPK3 in Cardiovascular Disease

Cardiovascular disease, a global health burden, is a class of disease that seriously affects quality of life and survival [85]. A genomic analysis verified the profile of circRNAs in the human heart and showed that the expression of circHIPK3 ranked third in healthy hearts [86]. A few studies explored the possible functions of circHIPK3 in the biology of cardiomyocytes after myocardial infarction (MI). Si et al. revealed circHIPK3 was strikingly expressed in fetal mice and silencing circHIPK3 significantly impeded proliferation of cardiomyocytes. Additionally, circHIPK3 was validated to accelerate angiogenic activities by enhancing the abilities to proliferate and form tubular structure of vessel endothelial cells. Furthermore, circHIPK3 might alleviate cardiac fibrosis by stabilizing and activating Notch1 intracellular domain (N1ICD), and then stimulate functions of endothelia cells by sponging miR-33a to facilitate connective tissue growth factor (CTGF) expression [87]. Wang and colleagues demonstrated that exosomal circHIPK3 of hypoxic cardiomyocytes dramatically expedited the proliferative and migrative abilities of cardiac endothelial cells, which was considered to be possibly conducive to minimizing the infarction region. Mechanistically, circHIPK3 depleted miR-29a and thereby promoted VEGFA expression [88]. However, Wu et al. obtained an opposite conclusion that knocking down circHIPK3 in MI cell models remarkably suppressed miR-93-5p expression to deactivate Rac1/PI3K/AKT signaling pathway, shrinking infarction area and protecting cardiomyocytes [89]. Concerning cardiac fibrosis, Liu et al. observed the level of circHIPK3 in cardiac fibroblasts was conspicuously raised under low oxygen circumstances. Overexpressed circHIPK3 strengthened TGF-β2 signaling pathway by hampering miR-152-3p so as to accelerate the process of fibrosis [90]. Liu et al. along with Ni and colleagues also demonstrated that silencing circHIPK3 contributed to retard cardio fibrosis triggered by angiotensin II [91,92]. Ni and colleagues further discovered the miRNA sponged by circHIPK3 as miR-29b-3p in their study [92]. CircHIPK3 was validated to be associated with ischemia-reperfusion (IR) injury of cardiomyocytes as well. Bai et al. found that by negatively regulating miR-124-3p, circHIPK3 overexpression decreased proliferation and stimulated death of cardiomyocytes in IR models [93]. Cheng and colleagues observed that under high-glucose conditions, circHIPK3 was downregulated by overexpressed miR-29a, and this interaction went downstream to AKT3/PIK3R3 pathway, enhancing the anti-apoptosis of cardiomyocytes and preventing cardiomyocytes from IR injury [94]. Some other studies focused on the role of circHIPK3 in hypoxia-reoxygenation (HR) injury of cardiomyocytes. Qiu et al. clarified that augmented circHIPK3 notably triggered autophagy and cell death of cardiomyocytes suffering HR injury. They also identified miR-20b-5p/autophagy-related 7 (ATG7) pathway modulated by circHIPK3 involved in IR injury of cardiomyocytes [95]. Exosomal circHIPK3 was considered by Wang et al. to have the potential to diminish oxidative injury of cardiac microvascular endothelial cells by abolishing miR-29a and thereby promoting expression of insulin-like growth factor-1 (IGF-1) [96]. Additionally, some studies reported new findings of circHIPK3 in pathophysiology of other cardiovascular diseases. For example, Wei et al. validated that circHIPK3 declined in atherosclerosis models and its overexpression showed pro-autophagy effects through sponging miR-190b, thereby relieving ATG7 [97]. Similarly, Zhang et al. enhanced the expression of circHIPK3 in atherosclerosis models and observed miR-106a-5p was negatively modulated when mitofusin 2 (MFN2) was stimulated. Intriguingly, calcium content was dramatically diminished in vascular smooth muscle cells (VSMC) [98]. Kang et al. knocked down circHIPK3 in VSMC and proliferative activities were consequently extenuated. Mechanistically, circHIPK3 hampered miR-637 expression and silenced CDK6 expression in VSMC [99]. Fan and colleagues illustrated that silencing circHIPK3 was able to mitigate myocarditis in mice and cell models caused by lipopolysaccharide [100]. Deng et al. verified that circHIPK3 could possibly be detrimental to patients with heart failure because it enhanced the function of adrenaline in the long term [101]. Xu et al. demonstrated that knocking down circHIPK3 in cardiac hypertrophy models preconditioned by transverse aortic constriction (TAC) and angiotensin II could improve the heart functions because it blocked miR-185-3p [102].

## 5. The Role of circHIPK3 in Diabetes Mellitus (DM) and Its Complications

DM is a major worldwide health issue caused by multiple elements such as genes, environment, and epigenetic factors [103]. Rezaeinejad et al. discovered that circHIPK3 level was highly elevated in Type 2 DM and circHIPK3 was associated with HbA1c, fasting blood glucose, etc. [104]. Liu and colleagues detected an overexpressed circHIPK3 in the kidney specimens of diabetic nephropathy mice and cell models. Mechanistically, circHIPK3 impeded miR-185 function and facilitated expression of cyclin D1, proliferating cell nuclear antigen (PCNA), TGF-β1, collagen Ⅰ, and fibronectin, and compounded diabetic nephropathy [105]. Wang et al. silenced aberrantly overexpressed circHIPK3 in diabetic cardiomyopathy models and found that the fibrosis of cardiomyocytes sharply decreased. Further, they identified miR-29b-3p as the target of circHIPK3 and Col1a1 as well as Col3a1, the downstream molecules [106]. CircHIPK3 was also reported by Cai et al. to be involved in hyperglycemia and insulin insensitivity via hampering miR-192-5p and stimulating transcription factor forkhead box O1 (FOXO1) [107]. Wang and colleagues clarified that overexpressed circHIPK3 silenced miR-124, escalating the degree of neuropathic pain induced by DM in mice [108]. Shan et al. verified that highly expressed circHIPK3 in retinal vascular silenced miR-30a-3p and strengthened the functions of vascular endothelial growth factor-C, Frizzled-4 (FZD4), and WNT2, aggravating the damage to vascular endothelial cells [109]. Wang et al. found an aberrantly elevated exosomal circHIPK3 in aortic endothelial cells of mice, strongly downregulating miR-106a-5p and stimulating expression of forkhead box O1 (Foxo1), which as a consequence expedited proliferation of vascular smooth muscle cells under high-glucose circumstances [110]. On the other hand, despite the harmful overexpression of circHIPK3 as mentioned above, several studies found the opposite answers. For instance, Cao et al. described a diminished level of circHIPK3 in the endothelial cells of the umbilical vein and primary aortic in human DM. CircHIPK3 depletion exacerbated the injury of endothelial cells by targeting miR-124 [111]. Zhuang et al. observed that the enhancement of circHIPK3 expression was capable of extenuating DM-induced renal tubular epithelial cell injury by decreasing miR-326 or miR-487a-3p to boost Sirtuin 1 (SIRT1) function [112]. Jiang et al. revealed that upregulation of circHIPK3 exerted its anti-apoptotic functions on cardiomyocytes by negatively mediating PTEN in diabetic cardiomyopathy [113].

## 6. The Role of circHIPK3 in Inflammatory Diseases

The pathophysiological mechanisms of different inflammatory diseases vary considerably. CircHIPK3 was confirmed to play a pivotal role in the occurrence of inflammation in a few human diseases. Wu et al. discovered that circHIPK3 level was remarkably lifted in cartilage specimens of human osteoarthritis, negatively targeting miR-124 and stimulating SRY-box transcription factor 8 (SOX8) expression. Consequently, the proliferation of chondrocytes was notably promoted in osteoarthritis [114]. Li et al. observed a similar signaling pathway but reached a different conclusion. Mechanistically, circHIPK3 originating from extracellular vesicles alleviated chondrocyte damage by blocking miR-124-3p to enrich the expression of myosin heavy-chain 9 (MYH9) [115]. In gouty arthritis, Lian and colleagues elucidated that circHIPK3 was enriched in mononuclear cells of synovial fluid, which strikingly repressed miR-561 and miR-192, and thereby facilitated pro-inflammatory functions of toll-like receptor 4 (TLR4) and NLR family pyrin domain containing 3 (NLRP3) [116]. Zhu et al. demonstrated that circHIPK3 expression obviously escalated along with lncGAS5 in allergic rhinitis mice models. Through silencing miR-495, the two molecules sparked Th2 differentiation and resulted in deterioration of allergic rhinitis [117]. In spinal cord injury (SCI), Yin et al. found that circHIPK3 was sharply diminished in SCI cell models and circHIPK3 overexpression attenuated inflammation and neuron death by downregulating miR-382-5p and then strengthened dual specificity phosphatase 1 (DUSP1) expression [118].

## 7. The Role of circHIPK3 in Other Human Diseases

### 7.1. Asthma

Airway remodeling plays an essential role in the occurrence and development of asthma [119]. Lin et al. detected an augmented expression of circHIPK3 in airway smooth muscle cells (ASMCs). In this condition, miR-326 was proved to be inhibited by circHIPK3 and its downstream factor stromal interaction molecule 1 (STIM1) was activated. As a consequence, proliferation and migration of ASMCs was accelerated, whereas apoptosis was retarded under this modulation [120]. Jiang and colleagues revealed another signaling pathway induced by circHIPK3 in which miR-375, as the target factor, was silenced by circHIPK3 and then the expression of matrix metallopeptidase 16 (MMP-16) was upregulated, resulting in escalation of proliferation and migration in ASMCs [121].

### 7.2. Osteoporosis

Osteoporosis is an age-related disease which seriously affects daily movement of the elderly [122]. Liang et al. showed that circHIPK3 decreased in osteoporosis models, and that enhancing expression of circHIPK3 significantly abolished miR-124 and prevented human osteoblasts from apoptosis caused by hydrogen peroxide [123]. Similarly, Zhu and colleagues reported that circHIPK3 overexpression had protective effects on osteoblasts injured by dexamethasone, in which miR-124 also participated [124].

### 7.3. Cataract

Cataract is a common disease of ophthalmology that has become a global burden in the past decades [125]. Cui et al. along with Liu et al. validated that circHIPK3 level declined in human lens epithelium cells (HLECs) and overexpressed circHIPK3 preserved the viability of HLECs. Mechanistically, the former verified that circHIPK3 overexpression absorbed miR-221-3p and stimulated PI3K/AKT pathway, whilst the latter demonstrated that elevated circHIPK3 diminished miR-193a and expedited alpha A crystallin (CRYAA) expression [126,127].

## 8. Conclusions

In parallel with the rapid advancement of RNA-sequencing and bioinformatics tools, scientists have gained deeper understanding of the specific roles that circRNAs play in human diseases. During the past decade, this has been particularly demonstrated in the gradually clarified biogenesis and expression of circRNAs in various organs and tissues, and the huge amount of circRNAs identified. Still, their regulatory network is rather complicated.

circHIPK3, a member of the circRNA family, has grabbed increasing attention from scientists. Accumulating studies are now trying to set forth the pathophysiological effects of circHIPK3 in a wide range of human diseases, including multiple malignancies, cardiovascular diseases, DM, inflammatory diseases, and others. Emerging evidence demonstrates that circHIPK3, the same as other circRNAs, mainly acts as the so-called “miRNA sponge”, and that the circHIPK3/miRNA/protein regulatory network is the basic modulation mode of circHIPK3. Figure 2 highlights the miRNAs that can be sponged by circHIPK3.

After comprehensively reviewing published reports, we summarized and integrated the role of circHIPK3 in human diseases. In the new concept of early diagnosis and early treatment of human cancers, the exploration of novel tumor biomarkers is of great importance. CircHIPK3 would be an ideal choice because it is abnormally expressed in an enormous range of different tumor cells. In other human diseases, as mentioned above, circHIPK3 can serve as the biomarker used for early warning of pathophysiological changes and prediction of prognosis. With the development of RNA detection technologies, we are optimistic that the detection of circHIPK3 will be applied in clinical practice. More importantly, the vast majority of preclinical experiments demonstrate the landscape of circHIPK3/miRNA/protein regulatory networks in a variety of human diseases. It has been verified that circHIPK3 plays a significant role in cell biology, not only in tumors but also in other diseases. Therefore, we reasonably predict that drugs targeting circHIPK3 may have a promising clinical application prospect and will further promote the individualized and precise treatment of human diseases in the future.

At the same time we noticed that some issues in this area still need further exploration. For instance, more efforts ought to be devoted to improving and enriching the regulatory mechanisms of circHIPK3, because contradictory conclusions have been reached by several similar studies regarding the expression and functions of circHIPK3 in diseases such as GC, osteosarcoma, and MI.

In conclusion, circHIPK3, despite its pending issues, has the potential to become a diagnostic biomarker and therapeutic target in clinical practice in the days to come.

## Figures and Tables

**Figure 1 diagnostics-12-02469-f001:**
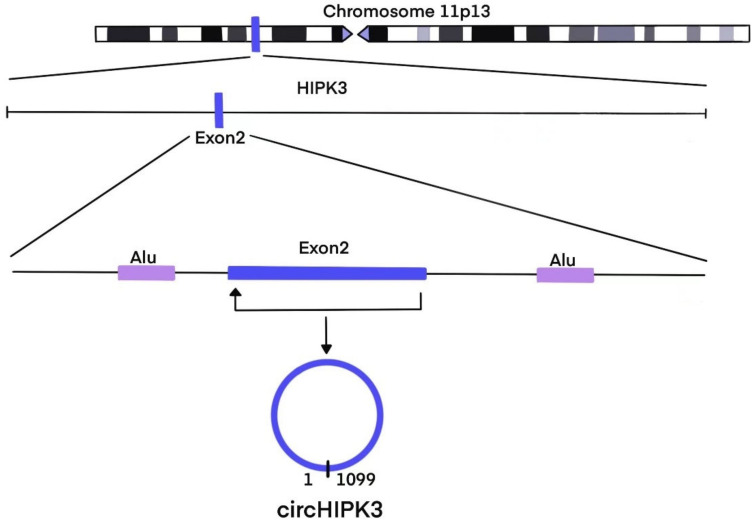
The biogenesis of circHIPK3.

**Figure 2 diagnostics-12-02469-f002:**
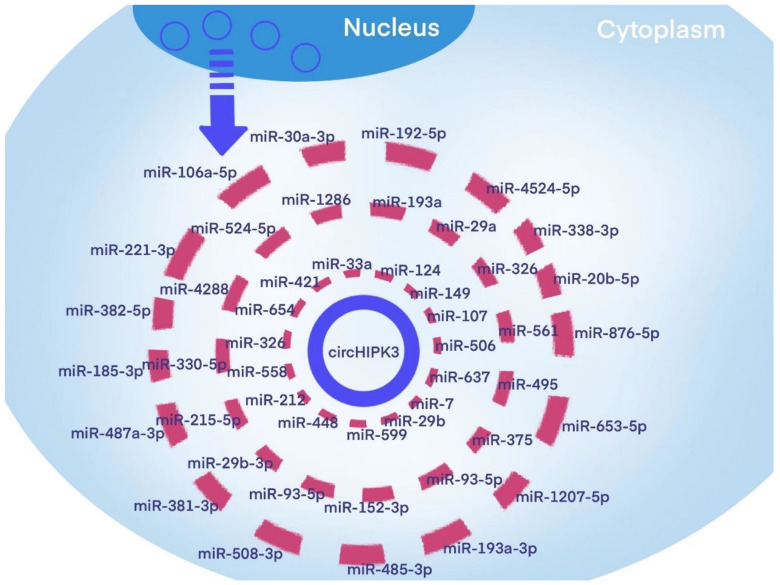
CircHIPK3 and the miRNAs that it sponges.

**Table 1 diagnostics-12-02469-t001:** The role of circHIPK3 in human cancers.

Cancer types	Location	Expression	Clinical Samples (n)	Cell Lines or Models	Animal Models	Signaling Pathway	Relevant Cell Biology	Relevant Clinical Features
lc	Cytoplasm	Upregulated	n = 15	A549 and BEAS-2B	-	miR-124/(SphK1, CDK4, STAT3)	Proliferation, migration and invasion	-
			-	A549, H1299, HCC-827, PC-9, H1975 and H838	-	miR-124-3p/STAT3	Proliferation, migration and invasion	-
			n = 25	16HBE, SPC-A1, A549, NCI-H1299, and NCI-H1650	-	miR-149/FOXM1	Proliferation and apoptosis	-
			-	H1299, A549, and BEAS-2B	-	miR-107/BDNF	Proliferation and migration	-
		Downregulated	n = 110	A549	-	-	Gefitinib resistance	Tumor size, TNM stage, lymph node metastasis
HCC	Cytoplasm	Upregulated	n = 50	Huh7, MHCC-LM3, HepG2, SMMC-7721, PLC, L02 and HEK293T	Nude mice	miR-124/AQP3	Proliferation and migration	HBV-DNA copy number, liver cirrhosis, tumor differentiation and TNM stage
			n = 30	HepG2, SMMC-7721, Bel-7402, Huh-7 and HL7702	BALB/c male nude mice	miR-124 or miR-506/PDK2	Proliferation and invasion	-
			-	HepG2, Hep3B, Huh-7, SKHep1 and THLE-3	BALB/C nude mice	miR-338-3p/ZEB2/EMT	Migration and invasion	Survival
			n = 19	SMMC-7721, Huh7, HepG2 and Hep3B	-	miR-124-3p and miR-4524-5p/MRP4	Migration and drug resistance	-
ESCC	Cytoplasm	Upregulated	n = 42	kyse-150, kyse-410, KYSE-510, ECA-109, EC-18, TE-13 and NE1	BALB/c female nude mice	miR-599/c-MYC	Proliferation and invasion	-
			n = 32	KYSE-150, KYSE-410, ECA-109 and KYSE180	BALB/c nude mice	miR-124/AKT3	Proliferation and migration	-
GC	Cytoplasm	Upregulated	n = 63	XGC-1, XGC-2, MGC-803 and BGC-823	-	miR-29b and miR-124	Proliferation and invasion	-
			n = 30	SGC-7901, MKN45, MGC-803, AGS, BGC-823, GES-1 and SGC-7901	BALB/c male nude mice	miR-107/BDNF	Proliferation and migration	-
			n = 26	GES-1, HGC-27 and AGS	BALB/c male nude mice	miR-876-5p/PIK3R1	Proliferation, migration and invasion	-
			n = 31	MGC803 and BGC823	-	miR-653-5p and miR-338-3p/NRP1/ERK/AKT	Promote metastasis	-
			n = 10	BGC-823, CRL-5822, SGC-7901, AGS and GES-1	-	miR-637/AKT1	Proliferation, migration and invasion	-
			n = 53	BGC, MGC, SGC, MKN and GES	-	Wnt/β-catenin	Proliferation and migration	Prognosis
		Downregulated	n = 30	-	-	-	-	Clinical stage, age
CRC	Cytoplasm	Upregulated	n = 178	FHC, HCT116, HT29, SW480, SW620,DLD1	BALB/c male nude mice	miR-7	Proliferation, migration, invasion and apoptosis	Pathological T category, lymph node metastasis, distant metastasis, TNM stage and prognosis
			n = 50	HT29, LOVO, SW480, PKO and NCM460	-	miR-1207-5p/FMNL2	Proliferation, migration and invasion	Prognosis
			n = 49	HT29, HCT116, HEK293T, 5FU-resistant and OXA-resistant cells	BALB/c male nude mice	miR-637/STAT3	Drug resistance	-
PCa	Cytoplasm	Upregulated	n = 26	RWPE-1, 22RV1, PC-3, DU145, and LNCaP	BALB/c male nude mice	miR-193a-3p/MCL1	Proliferation and invasion	Prognosis and tumor stage
			n = 60	RWPE-1, 22RV1, PC-3, DU145, and LNCaP	-	miR-338-3p/ADAM17	Proliferation and invasion	-
			n = 45	RWPE-1, 22RV1, PC-3, DU146, and LNCaP	-	miR-338-3p/Cdc2 and Cdc25B	Proliferation and G2/M transition	Gleason score
			n = 14	PC-3, VCaP, DU145, LNCaP, 22RV1 and PWPE-1	BALB/c male nude mice	miR-448/MTDH	Proliferation, migration and invasion	-
	Exosome	Upregulated	n = 35 (blood)	RWPE-1, 22RV1 and DU145	BALB/c male nude mice	miR-212/BMI-1	Proliferation, migration, invasion and apoptosis	-
RCC	Cytoplasm	Upregulated	n = 48	HK-2, Caki-1, ACHN, 786-O, 769-P and A498	BALB/c female nude mice	miR-485-3p/EMT	Proliferation, migration, invasion and apoptosis	-
			n = 50	A498, 786-O, 769-P and HRPTEpiC	-	miR-5083p/CXCL13	Proliferation, migration and invasion	-
			-	RPTECs	-	miR -381-3p/MRP4	Drug resistance	-
		Downregulated	n = 40	Caki-1, 786O, ACHN, A498 and HK-2	Nude mice	miR-637	Migration and invasion	-
Bladder cancer	Cytoplasm	Downregulated	n = 44	T24, UMUC3, SV-HUC-1, HUVEC	BALB/c female nude mice	miR-558/HPSE	Proliferation, migration, invasion, and angiogenesis	Lymph node metastasis
			n = 68	CCC-HB-2, SV-HUC-1, T24, J82 and UMUC3	-	-	-	Gemcitabine resistance, prognosis
			n = 457	RT4, HT1376, T24, FL3 and HCV29	-	-	-	Aggressiveness
Breast cancer	Cytoplasm	Upregulated	n = 48	MCF-10A, MCF7, SK-BR-3, BT549, BT20, MDA-MB-231	-	miR-326	Proliferation, migration and invasion	-
			n = 48	MCF-10A, MCF7, SK-BR-3, BT549, BT20, MDA-MB -231, MDA-MB-453	BALB/c male nude mice	miR-326	Proliferation, migration and invasion	-
			n = 37	MDA-MB-231, MCF-7, MDA-MB-231/PTX and MCF-7/PTX	BALB/c nude mice	miR-1286/HK2	Proliferation and paclitaxel resistance	-
			n = 50	MCF-10A, MCF-7, MDA-MB-231, MDA-MB-468 and MDA-MB-453	Nude mice	miR-193a/HMGB1/PI3K/AKT	Proliferation and invasion	-
	Exosome		-	MCF-7/Pr, MCF-7/Tr, SKBR3/Pr and SKBR3/Tr	BALB/c female nude mice	-	Trastuzumab chemoresistance	-
			-	MCF-7, MDA-MB-231, MDA-MB-453, SK-BR-3, BT-474, HUVEC	BALB/c nude mice	miR-124-3p/MTDH	Angiogenesis	-
OC	Cytoplasm	Upregulated	n = 69	HOEC, A2780, HO-8910, SKOV3 and CAOV3	-	-	-	Prognosis
			n = 21	A2780 and SKOV3	-	-	Proliferation, migration, invasion and apoptosis	-
CC	Cytoplasm	Upregulated	n = 70	SiHa and HeLa	-	miR-485-3p/FGF2	Proliferation, migration and invasion	-
			n = 45	HeLa, CaSki, SiHa, C-33A, C-4I, SW756 and End1/E6E7	-	miR-338-3p/HIF-1α/EMT	Proliferation, migration, invasion and apoptosis	-
Osteosarcoma	Cytoplasm	Upregulated	n = 10	U2OS and SW1353	-	miR-637/STAT3	Migration and invasion	-
			n = 12	hFOB 1.19, HOS, MG-63, U2OS and SJSA	-	miR-637/HDAC4	Proliferation, migration and invasion	-
		Downregulated	n = 82	SaoS2, HOS, KH-OS, MG63, 143B and U2-OS	-	-	Proliferation, migration and invasion	-
Glioma	Cytoplasm	Upregulated	n = 48	NHAs, U87 and U251	BALB/c female nude mice	miR-654/IGF2BP3	Proliferation, migration and invasion	Prognosis
			-	HEB, U87, U251, LN229 and LN308	-	miR-124-3p/STAT3	Proliferation, invasion, cell cycle and apoptosis	-
			Unknown	SW1783, and U373	-	miR-124/CCND2	Proliferation and invasion	-
			n = 80	A172 and U251	-	miR-524-5p/KIF2A	Proliferation, migration, invasion and apoptosis, temozolomide resistance	-
	Exosone		n = 56 (tumor and blood)	NHA, A172, U251, A172/TR and U251/TR	Nude mice	miR-421/ZIC5	Proliferation, migration, invasion and apoptosis, temozolomide resistance	-
OSCC	Cytoplasm	Upregulated	n = 30	HNOK, H357, SCC-15, SCC-4 and SCC-9	BALB/c female nude mice	miR-381-3p/YAP1	Proliferation, migration, invasion and apoptosis	-
			n = 40	NOK, OSCC-15, Tca8113, SCC-9, SCC-25, and HSC-2	C57BL/6 nude mice	miR-637/NUPR1/PI3K/AKT	Proliferation, invasion and apoptosis	Tumor size and histopathological grade
CML	Cytoplasm	Upregulated	n = 100	1D3, K562, KCL22, AR230-r, LAMA84-s and Kasumi-4	-	-	-	Prognosis
ALL	Cytoplasm	Upregulated	-	Mononuclear cells	-	-	-	-
NPC	Cytoplasm	Upregulated	n = 63	NP69, SUNE1, CNE1, CNE2, SUNE2, and 6-10B	BALB/c female nude mice	miR-4288/ELF3	Proliferation and invasion	-
Pancreatic cancer	Cytoplasm	Upregulated	n = 28	PANC-1 and SW 1990	BALB/c nude mice	miR-330-5p/RASSF1	Proliferation, migration, invasion, apoptosis and gemcitabine resistance	-
Thyroid cancer	Cytoplasm	Upregulated	n = 10	K1, CAL-62, TPC1, Nthy-ori 3-1	-	miR- 338-3p/RAB23	Proliferation, migration and invasion	-
GBC	Cytoplasm	Upregulated	n = 13	Mz-ChA-1, QBC939 and GBC-SD	-	miR-124/ROCK1/CDK6	Proliferation and apoptosis	-
Melanoma	Cytoplasm	Upregulated	n = 23	HEMa-LP, CHL-1 and A375	-	miR-215-5p/YY1	Proliferation and apoptosis	-
Glioblastoma	Extracellular vesicle	Upregulated	-.	-	-	-	-	-

Abbreviations: LC, lung cancer; SphK1, sphingosine kinase 1; CDK4, cyclin-dependent kinase 4; STAT3, signal transducer and activator of transcription 3; FOXM1, forkhead box M1; BDNF, brain-derived neurotrophic factor; HCC, hepatocellular carcinoma; AQP3, aquaporin 3; PDK2, pyruvate dehydrogenase kinase 2; MRP4, multidrug resistance protein 4; ZEB2, zinc finger E-box binding homeobox 2; EMT, epithelial-mesenchymal transition; ESCC, esophageal squamous cell carcinoma; AKT3, serine/threonine kinase 3; GC, gastric cancer; PIK3R1, phosphoinositide-3-kinase regulatory subunit 1; NRP1, neuropilin 1; ERK, extracellular signal-regulated kinase; AKT, protein kinase B; AKT1, serine/threonine kinase 1; CRC, colorectal cancer; FMNL2, formin-like 2; PCa, prostate cancer; MCL1, myeloid cell leukemia 1; ADAM17, a disintegrin and metalloproteinases 17; MTDH, metadherin; BMI-1, B-cell specific MMLV insertion site-1; RCC, renal cell carcinoma; CXCL13, chemokine ligand 13; HPSE, heparanase; HMGB1, high mobility group box-1; PI3K, phosphoinositide-3-kinase; HK2, hexokinase 2; OC, ovarian cancer; CC, cervical cancer; FGF2, fibroblast growth factor 2; HIF-1α, hypoxia-inducible factor-1alpha; HDAC4, histone deacetylase 4; IGF2BP3, insulin-like growth factor 2 mRNA-binding protein 3; CCND2, cyclin D2; ZIC5, zinc finger of the cerebellum 5; KIF2A, kinesin family member 2A; OSCC, oral squamous cell carcinoma; YAP1, Yes-associated protein1; NUPR1, nuclear protein 1; CML, chronic myeloid leukemia; ALL, acute lymphoblastic leukemia; NPC, nasopharyngeal cancer; ELF3, E74 like ETS transcription factor 3; RASSF1, RAS-Association domain family 1; RAB23, Ras-like in rat brain 23; GBC, gallbladder cancer; ROCK1, rho-associated protein kinase 1; CDK6, cyclin-dependent kinase 6; YY1, Yin Yang 1.

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
