# Peer review of "The Diagnostic and Therapeutic Role of Circular RNA HIPK3 in Human Diseases"

_diagnostics, 2022, doi:10.3390/diagnostics12102469_

Round 1

Reviewer 1 Report

Well done!

Author Response

Thank you very much for your comment!

Reviewer 2 Report

Feng etal., attempts to explore the diagnostic and pathological role of CirRNA HIPK3 in human diseases.

My concerns are as below.

1.       English language and grammatical errors are prevalent throughout the manuscript., such as in line 51 …activating sone other miRNAs … some other”.

2.       The authors described the biogenesis of circHIPK3. However how it’s transcription is regulated is missing.

3.       In this review, circHIPK3 s upregulated in most of the cancer. Can it be considered as “tumor marker”. Please discuss this point in the review.

4.       The review lacks the future perspective in terms of therapeutic role of circHIPK3.

Author Response

Thank you very much for your comments! We have made some improvements accordingly. Please see the revised manuscript.

Reviewer 3 Report

The paper by Yanfei Feng et al., entitled The Diagnostic and Therapeutic Role of Circular RNA HIPK3 in Human Diseases is a large review of the works that reported a potential role of HIPK3 in the pathogenesis and/or the course of many a tumor and non-tumor diseases.

The paper is well documented, well written, and deserves to be published. The reader is somewhat bewildered considering that the same molecular entity could be implied in so numerous and diverse pathologies. A hierarchy of the works according to the strength of the association between HIPK3 expression and disease might be helpful to reduce this apparent pleomorphic impact. Of note, the promotion of cell proliferation is provided as the most frequent Relevant Cell Biology factor (table 1). Might it be the common mechanism underlying the highly common involvement of HIPK3 in so various diseases? A comment of the authors on this point would be of interest.

Author Response

(The authors gave the same response as above.)
